# BayesGmed: An R-package for Bayesian causal mediation analysis

**Belay B. Yimer**[1]*, **Mark Lunt**[1], **Marcus Beasley**[2], **Gary J. Macfarlane**[2], **John McBeth**[1]

**1** Centre for Epidemiology Versus Arthritis, University of Manchester, Manchester, United Kingdom, **2** Aberdeen Centre for Arthritis and Musculoskeletal Health (Epidemiology Group), University of Aberdeen, Aberdeen, United Kingdom

* belaybirlie.yimer@manchester.ac.uk

**Data Availability Statement:** The Epidemiology Group, University of Aberdeen, are the owner of the dataset used in this paper and queries related to the data should be directed to epidemiology@abdn.

## Abstract

### Background

The past decade has seen an explosion of research in causal mediation analysis. However, most analytic tools developed so far rely on frequentist methods which may not be robust in the case of small sample sizes. In this paper, we propose a Bayesian approach for causal mediation analysis based on Bayesian g-formula, which will overcome the limitations of the frequentist methods.

### Methods

We created **BayesGmed**, an R-package for fitting Bayesian mediation models in R. The application of the methodology (and software tool) is demonstrated by a secondary analysis of data collected as part of the MUSICIAN study, a randomised controlled trial of remotely delivered cognitive behavioural therapy (tCBT) for people with chronic pain. We tested the hypothesis that the effect of tCBT would be mediated by improvements in active coping, passive coping, fear of movement and sleep problems. We then demonstrate the use of informative priors to conduct probabilistic sensitivity analysis around violations of causal identification assumptions.

### Result

The analysis of MUSICIAN data shows that tCBT has better-improved patients' self-perceived change in health status compared to treatment as usual (TAU). The adjusted log-odds of tCBT compared to TAU range from 1.491 (95% CI: 0.452–2.612) when adjusted for sleep problems to 2.264 (95% CI: 1.063–3.610) when adjusted for fear of movement. Higher scores of fear of movement (log-odds, -0.141 [95% CI: -0.245, -0.048]), passive coping (log-odds, -0.217 [95% CI: -0.351, -0.104]), and sleep problem (log-odds, -0.179 [95% CI: -0.291, -0.078]) leads to lower odds of a positive self-perceived change in health status. The result of **BayesGmed**, however, shows that none of the mediated effects are statistically significant. We compared **BayesGmed** with the **mediation** R- package, and the results were comparable. Finally, our sensitivity analysis using the **BayesGmed** tool shows that the

ac.uk. The data can be accessed upon a formal data sharing agreement.

**Funding:** The author(s) received no specific funding for this work.

**Competing interests:** The authors have declared that no competing interests exist.

direct and total effect of tCBT persists even for a large departure in the assumption of no unmeasured confounding.

## Conclusion

This paper comprehensively overviews causal mediation analysis and provides an open-source software package to fit Bayesian causal mediation models.

## 1. Introduction

Studies in the health and behavioural sciences often aim to understand whether and, if so, how an intervention causes an outcome. The randomised controlled trial is considered the most rigorous method for answering the "whether" question, but often the "how" part remains unclear. Causal mediation analysis plays an important role in understanding the mechanism by which an intervention produces changes in the outcome. Understanding how an intervention works can be key for further improvement and targeting of an intervention program.

There is a fast-growing methodological literature on causal mediation analysis [1–11]. One of the most important developments in mediation analysis is the incorporation of the causal inference approach or the potential outcomes framework (POF) to estimate causal mediation effects. This has led to (i) the formulation of different estimands (effect definitions) that have explicitly causal interpretations, (ii) clarification of the assumptions required for such effects to be estimated from observed data, (iii) a framework for conducting sensitivity analyses around violations of these assumptions, and (iv) has opened up a range of relevant estimation methods.

Within the POF, the regression-based [12] and the simulation-based [13] approaches are widely used for the estimation of causal mediation effects. The regression-based approach requires fitting parametric regressions models for the mediator and the outcome and involves approximations in the case of binary outcomes and mediators. On the other hand, the simulation-based approach is quite flexible and can accommodate parametric and non-parametric models. The regression-based approach implemented in SAS and SPSS macros relies on frequentist methods and the simulation-based approach implemented in the widely used **mediation** R package [14] is based on the quasi-Bayesian approximation where the posterior distribution of quantities of interest is approximated by their sampling distribution.

Recently, Bayesian modelling has been introduced to the mediation analysis literature [8, 15]. Compared to conventional frequentist mediation analysis, the Bayesian approach offers several advantages. First, Bayesian methods perform better when the sample size is small to moderate [15–17], which is particularly common in clinical trials. Second, it enables the straightforward construction of credible intervals for direct and indirect effects, providing a probabilistic interpretation of the uncertainty surrounding the estimated effects. Third, it offers the option of conducting a probabilistic sensitivity analysis where bias parameters that reflect the investigators' beliefs about unmeasured confounders are incorporated as prior information [18, 19]. However, the open-source software tools developed so far, such as **bmlm** [20] and **bayestestR** [21], have mainly focused on the Bayesian implementation of the product-method or linear structural equation modelling (LSEM) approach [22]. The LSEM framework has been criticised for its limited applicability beyond specific statistical models. Recently, Rix and Song, 2023 [23] introduced an R-package **bama**, which performs Bayesian mediation analysis based on the potential outcome framework. However, **bama** only handles continuous exposure and outcome. In this paper, we introduce a Bayesian estimation

procedure and open-source software tool, **BayesGmed**, for causal mediation analysis using the Bayesian g-formula approach. The proposed method follows the potential outcomes framework for effect definition and identification. We illustrate the applicability of the proposed method and software tool using data from MUSICIAN trial—a randomised controlled study [24].

## 2. Case study: MUSCIAN trial

To illustrate the methodology presented in this paper and demonstrate the use of the R-package **BayesGmed**, we used data from the MUSICIAN trial (Managing Unexplained Symptoms (CWP) In Primary Care: Involving Traditional and Accessible New Approaches (ClinicalTrials.gov Identifier: ISRCTN67013851)).

The MUSICIAN study was a 2x2 factorial trial that estimated the clinical effectiveness and cost-effectiveness of remotely (by telephone) delivered cognitive-behavioral therapy (tCBT), an exercise program, and a combined intervention of tCBT plus exercise, compared with treatment as usual (TAU) among people with CWP. For a complete discussion about the study and setting, we refer to [24]. Briefly, a total of 442 patients with CWP (meeting the American College of Rheumatology criteria) were randomised to one of the four treatment arms. The primary outcome was a 7-point patient global assessment scale of change in health since trial enrollment (range: 1: very much worse to 7: very much better) assessed at baseline and 6 months (intervention end) and 9 months after randomisation. A positive outcome was defined as "much better" or "very much better." Secondary outcomes including the Tampa Scale for Kinesiophobia (TSK) [25] (to measure fear of movement; score range, 17–68), the Vanderbilt Pain Management Inventory (VPMI) [26] (for assessing active and passive coping strategy use), and the Sleep Scale [27] (to measure sleep quality; score range, 0–20; higher scores indicate more sleep disturbance) were also assessed at baseline, 6 month and 9 months after randomisation.

Previous analysis of the MUSICIAN trial data has shown a significant benefit of tCBT in people with chronic pain as compared to treatment as TAU [24]. However, little is known about the mechanisms that lead to improvement. In this paper, using the MUSICIAN trial data, we aim to test the hypothesis that the effect of tCBT on the primary outcome is mediated by reductions in fear of movement, passive coping strategies, and sleep problems and an increase in the use of active coping strategies [Fig 1].

The analysis in this paper focuses on the outcome measured six months after randomisation and compares tCBT with treatment as usual. Baseline characteristics of the study cohort and outcome distribution at 6 months are presented in Table 1.

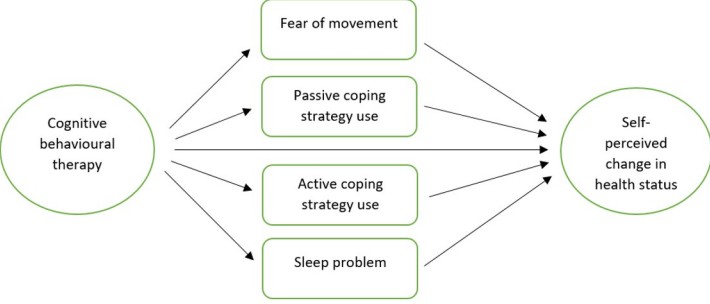

**Fig 1. Causal directed acyclic graph (DAG) for the MUSICIAN study.**

**Table 1. Baseline characteristics of study cohort and outcome at 6 months post-randomisation.**

| Characteristics | TAU | tCBT |
|---|---|---|
| **Baseline** | | |
| **N** | 109 | 112 |
| **Gender** | | |
| Female, n (%) | 76 (69.72) | 80 (71.42) |
| **Age, mean (SD)** | 56.4 (12.5) | 56.6 (13.7) |
| **Outcome at 6 month** | | |
| **Perceived health status since baseline** | | |
| Much better or very much better, n (%) | 7 (6.42) | 26 (23.21) |
| **Fear of movement (Kinesiophobia), mean (SD)** | 36.0 (6.75) | 34.2 (6.31) |
| **Active coping strategy use, mean (SD)** | 24.5 (4.50) | 25.4 (4.15) |
| **Passive coping strategy use, mean (SD)** | 28.0 (8.13) | 27.6 (7.60) |
| **Sleep problems, mean (SD)** | 9.96 (6.03) | 7.83 (5.61) |

## 3. The Mathematical framework for causal mediation analysis

In this section, we start by reviewing the ingredients of causal mediation analysis including definition of causal estimands/effects and the identification assumptions needed to learn those effects from observed data. We then describe how those causal estimands can be estimated from observed data using the Bayesian g-formula approach. To simplify our presentation, we restrict our examples to the context of an observed set of time-fixed variables.

### 3.1 Definition of causal mediation effects

The first step in causal mediation analysis is defining the causal effects of interest. We will start with the definition of the total treatment effect and then introduce the direct and indirect effects.

Consider estimation of the causal effect of a binary treatment assignment $A \in \{0, 1\}$ on some observed outcome $Y$, where 1 and 0 stand for the treatment and control conditions. Following the potential outcome framework concept [1], we denote the potential outcome that would have been observed for an individual had the exposure $A$ been set to the value $a$ by $Y(a)$. For the dichotomous treatment, we denote the outcome variable for the *ith* individual that would have been observed under the treatment value $\alpha = 1$ by $Y_i(1)$ and the outcome variable for the *ith* individual that would have been observed under the treatment value $\alpha = 0$ by $Y_i(0)$. Individual causal effects are defined as a contrast of the values of these two potential outcomes and treatment $A$ has a causal effect on an individual's outcome $Y$ if $Y_i(1) \neq Y_i(0)$. More formally, the total treatment effect at the individual level is defined on additive scale as $TE_i = Y_i(1) - Y_i(0)$. However, we never observe both potential outcomes for the same individual. What we observe is the realised outcome $Y_i$—the one corresponding to the treatment value experienced by the individual. Hence, identifying individual causal effects is generally not possible. However, under some assumptions to be discussed in the next subsection, the average total effect (ATE) in a population of individuals can be estimated from the observed data and it is defined as the average of the individual total effects over the population. That is, $ATE = \mathrm{E}[Y(1) - Y(0)]$. Put simply, the ATE is interpreted as the average difference in the outcome had everyone in the target population received treatment $A = 1$ rather than $A = 0$. If the outcome is binary (coded 0/1), this definition is equivalent to $ATE = P(Y(1) = 1) - P(Y(0) = 1)$, a risk difference. Further, given pre-exposure or pre-treatment assignment variables $\boldsymbol{Z}$, the conditional average total effect is given by $\mathrm{E}[Y(1) - Y(0)|\boldsymbol{Z}]$.

Mediation analysis moves beyond calculation of average total treatment effects and instead seeks to explain the effect of the exposure on the outcome. This is achieved by splitting the total treatment effect in to direct and indirect effects (Fig 2). By extending the previous notations to a joint exposure $(A, M)$ with $M$ being the potential mediator, definition of direct and indirect effects can be constructed as follows.

Let $M_i(a)$ denote the potential value of a mediator of interest under the treatment status $A = a$ and let $Y_i(a, m)$ represent the potential outcome values under regime $A = a$ when the mediator $M$ is set to the value it would naturally take under either $A = a$. For a dichotomous exposure, the average controlled direct effect for mediator at level m given covariate **Z** is given by [1–3]

$$CDE\,(m) = \mathrm{E}[Y(1, \mathrm{m}) - Y(0, \mathrm{m})|\mathbf{Z}]. \tag{1}$$

The controlled direct effect expresses the exposure effect that would be realised if the mediator were controlled, i.e., set to a specific level for everyone. Controlled direct effects are relevant quantities when interest lies in the evaluation of an intervention that can shift or fix the mediator across the population. However, the controlled direct effect does not usually lead to the splitting of the total effect in to direct and indirect effect. That is, the total effect minus the controlled direct effect may not have the interpretation of indirect effect in situations where the direct effect is different at different levels of the mediator. Hence, we introduce below two additional quantities that can split the total effect in to direct and indirect effect. They are the natural direct and natural indirect effects.

The average natural direct and indirect effects, given a pre-exposure covariates Z, are defined as [1–3]

$$NDE(a) = \mathrm{E}[Y(1, \mathrm{M}(a)) - Y(0, \mathrm{M}(a))|\mathbf{Z}] \tag{2}$$

and

$$NIE(a) = \mathrm{E}[Y(a, \mathrm{M}(1)) - Y(a, \mathrm{M}(0))|\mathbf{Z}]. \tag{3}$$

The indirect effect $NIE$ represents the causal effect of the treatment on the outcome that can be attributed to the treatment-induced change in the mediator and the direct effect $NDE$ denotes the causal effect of the treatment on the outcome that can be attributed to causal mechanisms other than the one represented by the mediator, and their sum leads to the total effect. That is, $TE = NIE(1) + NDE(0) = NIE(0) + NDE(1)$. Note that, $NIE(1)$ and $NIE(0)$ may not be identical and a similar inequality holds for $NDE(1)$ and $NDE(0)$.

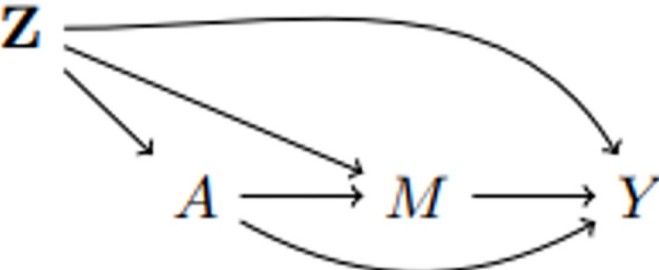

**Fig 2. Mediation with a single mediator M, exposure A, outcome Y, and confounders Z.**

## 3.2 Identification assumptions

To be able to identify or estimate the causal effects defined in 3.1, we need to rely on a set of assumptions. To estimate the above causal estimands from the observed data and ensure they have a causal interpretation, the following four conditions need to hold:

- *IA*1: $Y(a, m) \perp A|Z$: no-unmeasured confounder for the exposure-outcome relationship given the pre-exposure covariate Z.

- *IA*2: $Y(a, m) \perp M|A$, Z: no-unmeasured confounder for the mediator-outcome relationship given the pre-exposure covariate Z and the exposure A.

- *IA*3: $M(a) \perp A|Z$: no-unmeasured confounder for the exposure-mediator relationship given the pre-exposure covariate Z.

- *IA*4: $Y(a, m) \perp M(a^*)|Z$ for any value of *a*, *a\**, *and m*: no-measured or unmeasured confounder for the mediator-outcome relationship that is also influenced by the exposure.

Under assumptions IA1-IA4, the natural direct and indirect effects can be identified [2, 3, 5] by

NDE:

$$E\big[Y\big(a, M(a')\big) - Y\big(a', M(a')\big)|\mathbf{Z}\big] =$$

$$\int \int \{E[Y_i|M_i = m, A_i = a, \mathbf{Z}_i = \mathbf{z}] - E[Y_i|M_i = m, A_i = a', \mathbf{Z}_i = \mathbf{z}]\} dF_{M_i|A_i=a, \mathbf{Z}_i=\mathbf{z}}(m) dF_{Z_i}(\mathbf{z}). \quad (4)$$

and

NIE:

$$E\big[Y\big(a, M(a)\big) - Y\big(a, M(a')\big)|\mathbf{Z}\big] =$$

$$\int \int E[Y_i|M_i = m, A_i = a, \mathbf{Z}_i = \mathbf{z}]\Big\{ dF_{M_i|A_i=a, \mathbf{Z}_i=\mathbf{z}}(m) - dF_{M_i|A_i=a', \mathbf{Z}_i=\mathbf{z}}(m) \Big\} dF_{Z_i}(\mathbf{z}). \quad (5)$$

If the mediator is discrete, the integrals will be replaced by summation over the possible values of *M*. In the epidemiological literature, computation of causal effects using the above expression is called standardisation—a special case of g-computation.

Note that, to identify the control direct effect, only assumption IA1 and IA2 are need to hold. If assumption *IA*1 and *IA*2 hold, then the controlled direct effects are identified [2] by

$$E[Y(1, m) - Y(0, m)|\mathbf{Z}] = E[Y|A = 1, M = m, \mathbf{Z}] - E[Y|A = 0, M = m, \mathbf{Z}], \quad (6)$$

and the average controlled direct effect can be estimated from the data by averaging over the distribution of **Z**.

## 3.3 Estimation

After defining the causal estimands and specifying the necessary conditions for the estimand to be identified, the next step is doing the actual estimation from the observed data. In this section, we will introduce Bayesian modeling for causal effect estimation. Bayesian causal mediation analysis combines Bayesian modeling with the identification assumptions discussed in 3.2 to compute a posterior distribution over the causal estimands of interest.

Suppose we observe data $D = \{Y_i, M_i, A_i, \mathbf{Z}_i\}_{i = 1:n}$ on *n* independent individuals, where $A_i \in \{0, 1\}$ is a binary treatment indicator, $\mathbf{Z}_i$ is a vector of confounders, $M_i$ is a scalar candidate mediator, and $Y_i$ is a binary outcome of interest. Assume assumption $IA1 - IA4$ hold, and that

and that the following regression models for $Y$ and $M$ are correctly specified:

$$\text{logit}(P(Y_i = 1 | A_i, M_i, \mathbf{Z}_i)) = \alpha_0 + \boldsymbol{\alpha}'_Z \mathbf{Z}_i + \alpha_A A_i + \alpha_M M_i, \tag{7}$$

$$E[M_i | (A_i, \mathbf{Z}_i)] = \beta_0 + \boldsymbol{\beta}'_Z \mathbf{Z}_i + \beta_A A_i, \text{ with } \epsilon_i \sim N(0, \sigma^2). \tag{8}$$

In addition to the probability model for the conditional distribution of the outcome and the mediator (the likelihood), Bayesian inference requires a probability distribution over the unknown parameter vector, $\boldsymbol{\theta} = (\alpha_0, \boldsymbol{\alpha}_Z, \alpha_A, \alpha_M, \beta_0, \boldsymbol{\beta}_Z, \beta_A)$, governing this conditional distribution (i.e. a prior). Inference then follows from making probability statements about $\boldsymbol{\theta}$ having conditioned on the observed data (via the posterior). One of the key advantages of Bayesian inference is using priors one can obtain a stabilised causal effect estimates when data are sparse. Specification of priors to induce shrinkage is beyond the scope of this paper and we refer interested readers to[28]. For now, we assume suitable priors in line with the specific problem one is addressing are specified.

Bayesian estimation of causal effects rely on Bayesian analog of the g-formula (standardisation) and bootstrap estimation of the confounder distribution. The Bayesian analog to the g-formula [29] formulates the distribution of the counterfactual $Y_a$ as a posterior predictive value, integrating over the parameters $\boldsymbol{\theta}$ as well as the confounder distribution.

$$p(\tilde{y}(a)|o) = \int \int p(\tilde{y}|a, \tilde{\mathbf{z}}, \theta) p(\tilde{\mathbf{z}}|\boldsymbol{\theta}) p(\boldsymbol{\theta}|o) d\boldsymbol{\theta} d\tilde{\mathbf{z}}.$$

Integration over the parameters and the confounder distribution as well as the computation of causal effects involve the following 5 steps.

1. Given $B$ iterations, at the $b^{th}$ iteration obtain the posterior draws of the parameters $\boldsymbol{\theta}$ and denote them by $\boldsymbol{\theta}^{(b)} = (\alpha_0^{(b)}, \boldsymbol{\alpha}_z^{(b)}, \alpha_A^{(b)}, \alpha_M^{(b)}, \beta_0^{(b)}, \boldsymbol{\beta}_Z^{(b)}, \beta_A^{(b)})$.

2. Using the classical bootstrap, sample $n$ new values of $\mathbf{Z}$ with replacement from the observed $\mathbf{Z}$ distribution during iteration $b$ of the Markov Chain Monte Carlo and denote these resampled values as $\mathbf{Z}^{(1,b)}, \ldots, \mathbf{Z}^{(n,b)}$.

3. Get the potential outcome values

   a. Simulate the potential values of the mediator. Using the resampling of $\mathbf{Z}$ as described earlier, we can draw samples from the distributions of the counterfactuals $M(a)$ for $a \in \{0, 1\}$. At the $b^{th}$ MCMC iteration and for $i = 1, \ldots, n$,

   $$M(a)^{(i,b)} \sim \text{Normal}\big(\beta_0^{(b)} + \boldsymbol{\beta}_Z^{(b)} \mathbf{Z}^{(i,b)} + \beta_A^{(b)} a, \sigma^{(b)}\big)$$

   b. Given the potential value for the mediator, simulate the potential value for the outcome. For example, $Y(a, M(a)^{(i,b)})^{(i,b)}$ is simulated using

   $$Y(a, M(a)^{(i,b)})^{(i,b)} \sim \text{Bernoulli}\left(\text{logit}^{-1}\left(\beta_0^{(b)} + \boldsymbol{\beta}_Z^{(b)} \mathbf{Z}^{(i,b)} + \beta_A^{(b)} a + \alpha_M^{(b)} * M(a)^{(i,b)}\right)\right)$$

4. Compute draw of the causal effect estimates.

   a. NDE: $NDE(a)^{(b)} = \frac{1}{n} \sum_{i=1}^{n} \left\{ Y\left(a\prime, M(a\prime)^{(i,b)}\right)^{(i,b)} - Y\left(a, M(a')^{(i,b)}\right)^{(i,b)} \right\}$

   b. NIE: $NIE(a)^{(b)} = \frac{1}{n} \sum_{i=1}^{n} \left\{ Y\left(a, M(a)^{(i,b)}\right)^{(i,b)} - Y\left(a, M(a')^{(i,b)}\right)^{(i,b)} \right\}$

5. Get summary of causal effect estimates by taking the mean and quantiles of the causal effect estimates draws.

### 3.4 Sensitivity analysis

As described in section 3.1, estimating direct and indirect effects from observed data requires a series of assumptions. As a result, the main challenge in mediation analysis has been understanding bias from unmeasured confounding variables. Several methods have been proposed in the literature to explore the sensitivity of causal effect estimates to unmeasured confounding [5, 30, 31]. In our Bayesian causal mediation analysis R-package, presented in the following section, we implemented the Bayesian sensitivity analysis (BSA) proposed by [18]. BSA works by incorporating uncertainty about unmeasured confounding in the outcome and mediator model through a prior distribution. That is, we extend the outcome and mediator model in Eqs (8) and (9) to a triple set of structural equations

$$\text{logit}(P(Y_i = 1 | A_i, M_i, \mathbf{Z}_i)) = \alpha_0 + \boldsymbol{\alpha}'_Z \mathbf{Z}_i + \alpha_A A_i + \alpha_M M_i + \alpha_U U_i, \tag{9}$$

$$E[M_i | (A_i, \mathbf{Z}_i)] = \beta_0 + \boldsymbol{\beta}'_Z \mathbf{Z}_i + \beta_A A_i + \beta_U U_i, \text{ with } \epsilon_i \sim N(0, \sigma^2), \tag{10}$$

$$\text{logit}(P(U_i = 1 | A_i, \mathbf{Z}_i)) = \gamma_0 + \gamma_A A_i, \tag{11}$$

where the binary random variable $U$ that takes values 1 or 0 indicates the presence or absence of an unmeasured confounder and the parameters $\alpha_U$ and $\beta_U$ governs the association between $U$ and Y and U and $M$, respectively. Finally, $\gamma_0$ and $\gamma_A$ controls the prevalence of the unmeasured confounder within levels of the exposure variable A given Z.

The BSA approach proceeds by assuming a uniform prior distribution, *Uniform*$(-\delta, \delta)$, for the bias parameters $\alpha_U, \beta_U, \gamma_0$ and $\gamma_A$ where $\delta$ to represent the size of unmeasured confounding (E.g. $\delta = 0$ means no unmeasured confounding) [18]. The elicitation of $\delta$ (bias parameter), can be based on the investigators prior belief about the magnitude and direction of unmeasured confounding. In the absence of prior information on the direction or magnitude of unmeasured confounder, one can let the bias parameter to vary between zero and one, enabling an evaluation of continuous departures from the assumption of no unmeasured confounding assumption. For detailed discussion on the choice of informative priors for the sensitivity analysis we refer to McCandless [18]. To aid convergence, one may replace the *Uniform*$(-\delta, \delta)$ a Normal prior distribution. The estimation of direct and indirect effect using Eqs 9–11 follows the same procedure as described in section 3.3 but the potential outcome and mediator values now will also depend on the values of U. This way, the posterior distribution for the causal effect estimates incorporates uncertainty from bias (systematic error) in addition to uncertainty from random sampling (random error).

### 4. Implementation

The **BayesGmed** package implements Bayesian causal mediation analysis procedure described in the previous section in R using the probabilistic programming language **Stan** [32]. The latest development version of the R-package, **BayesGmed**, can be installed from GitHub via:

```
devtools::install_github("belayb/BayesGmed")
```

Models are fitted in **BayesGmed** using the following procedure:

```
bayesgmed(outcome, mediator, treat, covariates = NULL,
          dist.y = "continuous", dist.m = "continuous",
          link.y = "identity", link.m = "identity", data,
          control.value = 0, treat.value = 1, priors = NULL, ...
          )
```

The **BayesGmed** R-package currently handles continuous outcome—continuous mediator, binary outcome—binary mediator, continuous outcome—binary mediator, and binary outcome—continuous mediator. Currently, a multinormal, *MVN*(*location*, *scale*), prior is assigned to all regression parameters where the location and scale parameters are fixed to the following default values. The user can change the location and scale parameters by passing the location and scale parameters of the priors as a list as below

```
priors <- list(scale_m = 2.5*diag(P+1),
               scale_y = 2.5*diag(P+2),
               location_m = rep(0, P+1),
               location_y = rep(0, P+2),
               scale_sd_y = 2.5,
               scale_sd_m = 2.5)
```

where *P* is the number of covariates (including the intercept) in the mediator/ outcome model. For the residual standard deviation, a half-normal prior is assumed with mean zero. The user can change the scale_sd values as above.

To conduct sensitivity analysis, the *bayesgmed_sens* function in **BayesGmed** as follow:

```
bayesgmed_sens(outcome, mediator, treat, covariates = NULL,
               dist.y = "continuous", dist.m = "continuous",
               link.y = "identity", link.m = "identity", data,
               control.value = 0, treat.value = 1, priors = NULL, ...
               )
```

The *bayesgmed_sens* function have the same structure as the main function *bayesgmed* except one has to provide a list of priors for the bias parameters. Detailed vignettes describing the step-by-step use of **BayesGmed** to conduct causal mediation analysis on various types of outcomes and mediators are currently available at https://github.com/belayb/BayesGmed.

## 5. Results

We analysed the MUSICIAN trial data using the Bayesian causal mediation analysis framework presented in the previous section and implemented in the R-package **BayesGmed**. We investigated the potential mediating effect of each of the mediators separately, assuming independence between the mediators. We considered a logistic regression model for the outcome and a linear regression model for the mediator model (see S1 Text). For all model parameters, we assumed non-informative priors listed in S1 Text. We ran 4 Markov chain cycles, each with 4000 samples after 4000 burn-in samples and assessed convergence using standard MCMC convergence checks. For a simple comparison of the **BayesGmed** result with the result of the well-established method, we also analysed the data using the **mediation** R-package and presented the results side by side.

Compared to TAU, we found that tCBT has a significant positive effect on self-perceived change in health status (Table 2). The adjusted log-odds of tCBT on self-perceived change in health status compared to TAU range from 1.491 (95% CI: 0.452–2.612) when adjusted for sleep problems to 2.264 (95% CI: 1.063–3.610) when adjusted for fear of movement. Adjusted

**Table 2. MUSICIAN trial: Mediation analysis with one mediator at a time approach.** The Total effect, the average causal direct (ADE) and indirect effects (ACME) are presented in risk difference scale. The coefficients in the outcome model are in log odds scale and the coefficients of the mediator model are on a linear scale. All models are adjusted for age, sex and baseline GHQ median scores.

| | | Mediators | | | |
|---|---|---|---|---|---|
| | | Fear of movement | Active coping | Passive coping | Sleep problems |
| Analysis using the **BayesgMed** R-package | Outcome Model | | | | |
| | tCBT | **2.264 (1.063, 3.610)** | **1.180 (0.614, 3.305)** | **1.765 (0.272, 3.433)** | **1.491 (0.452, 2.612)** |
| | Mediator* | **-0.141 (-0.245, -0.048)** | -0.028 (-0.170, 0.116) | **-0.217 (-0.351, -0.104)** | **-0.179 (-0.291, -0.078)** |
| | Mediator Model | | | | |
| | tCBT | -1.776 (-3.815, 0.369) | 0.516 (-0.982, 2.053) | -0.546 (-3.076, 1.981) | **-2.350 (-4.132, -0.569)** |
| | Direct & indirect effects | | | | |
| | ADE (control) | **0.211 (0.085, 0.348)** | **0.156 (0.036, 0.288)** | **0.110 (0.007, 0.226)** | **0.151 (0.026, 0.288)** |
| | ADE (treated) | **0.233 (0.092, 0.376)** | **0.155 (0.036, 0.288)** | **0.115 (0.007, 0.234)** | **0.178 (0.039, 0.327)** |
| | ACME (control) | 0.014 (-0.050, 0.085) | -0.001 (-0.050, 0.050) | 0.006 (-0.051, 0.073) | 0.033 (-0.039, 0.111) |
| | ACME (treated) | 0.035 (-0.064, 0.142) | -0.001 (-0.094, 0.086) | 0.011 (-0.088, 0.109) | 0.060 (-0.046, 0.170) |
| | Total effect | **0.247 (0.106, 0.390)** | **0.154 (0.029, 0.288)** | **0.121 (0.000, 0.248))** | **0.211 (0.072, 0.353)** |
| | ADE (average) | **0.222 (0.099, 0.348)** | **0.155 (0.047, 0.277)** | **0.112 (0.015, 0.219)** | **0.165 (0.042, 0.294)** |
| | ACME (average) | 0.025 (-0.039, 0.096) | -0.001 (-0.054, 0.050) | 0.009 (-0.055, 0.077) | 0.046 (-0.026, 0.121) |
| Analysis using the **Mediation** R-package | ADE (control) | **0.201 (0.079, 0.350)** | **0.156 (0.060, 0.260)** | **0.109 (0.012, 0.230)** | **0.148 (0.051, 0.260)** |
| | ADE (treated) | **0.221 (0.091, 0.400)** | **0.155 (0.060, 0.270)** | **0.115 (0.015, 0.250)** | **0.173 (0.062, 0.290)** |
| | ACME (control) | 0.016 (-0.001, 0.050) | -0.001 (-0.010, 0.010) | 0.006 (-0.021, 0.040) | **0.032 (0.007, 0.070)** |
| | ACME (treated) | 0.036 (-0.001, 0.100) | -0.002 (-0.021, 0.020) | 0.011 (-0.033, 0.060) | **0.056 (0.014, 0.120)** |
| | Total effect | **0.237 (0.112, 0.410)** | **0.154 (0.062, 0.270)** | **0.121 (0.020, 0.270)** | **0.204 (0.088, 0.320)** |
| | ADE (average) | **0.211 (0.086, 0.380)** | **0.156 (0.060, 0.270)** | **0.112 (0.014, 0.240)** | **0.160 (0.056, 0.270)** |
| | ACME (average) | 0.026 (-0.001, 0.070) | -0.001 (-0.175, 0.090) | 0.047 (-0.651, 0.470) | **0.044 (0.011, 0.100)** |

for the intervention, the result of the outcome model revealed a significant relationship between self-perceived change in health status and fear of movement, passive coping, and sleep problem. Higher scores of fear of movement, passive coping, and sleep problem leads to lower odds of a positive self-perceived change in health status. However, the result of the mediator model shows that tCBT has a significant influence only on reducing sleep problem score (-2.350, 95% CI: -4.132, -0.569). tCBT had a negative relationship with fear of movement and passive coping score and a positive relationship with the active coping score but none of them are statistically significant.

The result of **BayesGmed** shows that none of the mediated effects are statistically significant, indicating that either the effect of tCBT on self-perceived change in health status is through other mechanisms independent of fear of movement, the use of active or passive coping strategies, and sleep problems or the study is too small to detect a significant mediated effect. The result of **BayesGmed** is comparable to the **mediation** R- package results except for the indirect effect estimates of sleep problems. Analysis using the **mediation** R-package shows a significant mediating effect of sleep problems. This is due to the relatively larger standard errors from **BayesGmed** since it accounts additional sources of uncertainty in the parameter estimation.

We applied BSA to the MUSICIAN trial data in order to explore sensitivity of the results to bias from unmeasured confounding. We considered three values for the bias parameter (i.e., $\gamma = (\gamma_0, \gamma_A, \beta_U, \alpha_U) \sim MVN(0, \delta I4)$, where $\delta = 0, 0.5,$ and 1) to denote varying level of departure from no unmeasured confounder assumption. When $\delta = 0$, we fit a model without unmeasured confounder. The results of BSA are presented in Fig 3. For brevity, we only presented

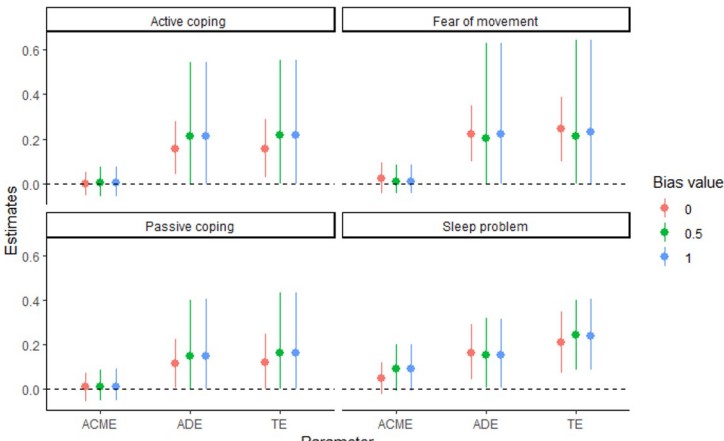

**Fig 3. MUSICIAN trial: Bayesian sensitivity analysis for varying levels of departure from no-unmeasured confounder assumptions.**

the results of the average direct (ADE), average indirect effect (ACME) and total effect (TE). Overall, BSA leads to a much wider credible intervals for all effects of interest than the Naive ($\delta = 0$). If we consider 95% credible interval overlap with zero in order to identify non-zero natural direct and indirect effects, then Fig 3 shows that the direct and total effect of cognitive behavioral therapy on changes in perceived health status persists even for a large departure in the assumption of no unmeasured confounding.

## 6. Concluding remark

In this paper, we introduced a Bayesian estimation algorithm for causal mediation analysis. We also provide an easy-to-use R-package for conducting Bayesian causal mediation analysis and assessing sensitivity of results for unmeasured confounder. Compared to the existing open-source tools for mediation analysis, **BayesGmed** has several advantages. First, point and interval estimates can be easily constructed for causal risk ratios, odds ratios, and risk differences by post-processing posterior draws from the fitted model. Second, priors can be specified to obtain more stabilised causal effect estimates than the frequentist procedure. Third, priors can also be used to conduct probabilistic sensitivity analyses around violations of key causal identification assumptions.

Using the proposed methodology, we analysed data from a randomised control trial with the aim of identifying mediators of tCBT on self-perceived change in health status in patients with chronic widespread pain. We showed the beneficial effect of tCBT compared to TAU, similar to previous reports [24]. However, none of the considered potential mediators (i.e. reduction in fear of movement, reduction in passive coping, reduction in sleep problem, and an increase in activing coping) were found to mediate the effect of tCBT. Except active coping, all of the potential mediating factors were found to have a statistically significant effect on the outcome of interest, but tCBT had a significant effect only on reducing sleep problems leading to a non-significant indirect effect. These results suggest that either improving the scope of tCBT or combining tCBT with other interventions that can target fear of movement, passive coping, and sleep problem would increase patient benefit. However, it is important to note that the MUSICIAN trial was not powered to detect mediators of the effect of tCBT on outcome. tCBT was associated with change in scores for fear of movement, active coping, passive

coping, and sleep problems in the expected direction, and the magnitude of effect was greatest for sleep problems. Whether these would mediate the effect of tCBT in an adequately powered trial remains unknown. However, the methods presented here would be able to address that question in an well-powered study. It also remains possible that tCBT exerts its influence through some other mechanism(s). It would be of interest to explore non-specific effects in non-blinded trials such as MUSICIAN.

To provide a simple comparison, we also conducted an analysis of MUSICIAN data using the **mediation** R-package alongside **BayesGmed**. The results obtained with both methods led to similar indirect effect estimates, except for the mediating effect of sleep problems, where the effect was found to be significant in the **mediation** R-package but not in **BayesGmed**. This discrepancy may be attributed to the small observed mediated effect in our study. As demonstrated by Yuan and MacKinnon, 2009 [16] through a simulation study, the Bayesian approach exhibits better 95% coverage than the frequentist approach when the sample size and the mediated effect is small. However, we recognize the need for a comprehensive simulation study to delve deeper into the comparison between **BayesGmed** and the **mediation** R-package.

At present, there are some limitations of the package **BayesGmed**. First of all, we assumed a parametric specification for the outcome and mediator model. In some situations, parametric models might be restrictive and a general non-parametric models might be preferred. Second, we only considered the case of single mediator and assumed no exposure mediator interaction. The Bayesian estimation algorithm we presented is quite generic and can easily be extended to accommodate the aforementioned limitation and we aim to extend the **BayesGmed** package to handle the above settings in a future version. Since the package is distributed as an open source software users can also update the package for their own needs.

## Supporting information

**S1 Text.**
(DOCX)

## Author Contributions

**Conceptualization:** Belay B. Yimer, Mark Lunt, John McBeth.

**Data curation:** Belay B. Yimer, Marcus Beasley.

**Formal analysis:** Belay B. Yimer.

**Methodology:** Belay B. Yimer, Mark Lunt, John McBeth.

**Project administration:** Belay B. Yimer.

**Resources:** John McBeth.

**Software:** Belay B. Yimer.

**Supervision:** Mark Lunt, Gary J. Macfarlane, John McBeth.

**Validation:** Belay B. Yimer, John McBeth.

**Visualization:** Belay B. Yimer.

**Writing – original draft:** Belay B. Yimer, Mark Lunt, John McBeth.

**Writing – review & editing:** Belay B. Yimer, Mark Lunt, Marcus Beasley, Gary J. Macfarlane, John McBeth.

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
