## [Decision Letter · Decision Letter 0]

6 Apr 2023

PONE-D-23-00922BayesGmed: An R-package for Bayesian Causal Mediation AnalysisPLOS ONE

Dear Dr. Yimer,

Thank you for submitting your manuscript to PLOS ONE. After careful consideration, we feel that it has merit but does not fully meet PLOS ONE’s publication criteria as it currently stands. Therefore, we invite you to submit a revised version of the manuscript that addresses the points raised during the review process.

We look forward to receiving your revised manuscript.

Kind regards,

Debo Cheng

Academic Editor

PLOS ONE

Journal Requirements:

"This research was supported by the Centre for Epidemiology Versus Arthritis (grant number 21755). The MUSICIAN trail, used as a case study in this paper, is funded by versus Arthritis (grant number 20748)."

Additional Editor Comments:

Reviewers find the idea interesting for causal mediation analysis, but also identify issues which should be addressed in the revision, such as more literature on mediation analysis and sensitivity analysis.

Reviewers' comments:

Reviewer's Responses to Questions

**Comments to the Author**

1. Is the manuscript technically sound, and do the data support the conclusions?

Reviewer #1: Yes

Reviewer #2: Partly

2. Has the statistical analysis been performed appropriately and rigorously? 

Reviewer #1: Yes

Reviewer #2: Yes

3. Have the authors made all data underlying the findings in their manuscript fully available?

Reviewer #1: Yes

Reviewer #2: Yes

4. Is the manuscript presented in an intelligible fashion and written in standard English?

Reviewer #1: Yes

Reviewer #2: Yes

5. Review Comments to the Author

Reviewer #1: This paper focuses on an important research problem, i.e., mediation analysis. The goal of many studies in health is to understand how interventions produce changes in outcomes. While RCTs are useful for determining whether an intervention causes an outcome, they may not provide information on how the intervention works. Causal mediation analysis can help answer the "how" question by identifying the mechanisms by which interventions affect outcomes. The potential outcomes framework (POF) is commonly used for causal mediation analysis and has led to the development of different effect definitions, assumptions for estimating these effects, and methods for conducting sensitivity analyses. Two widely used approaches for estimating causal mediation effects are the regression-based approach and the simulation-based approach. While Bayesian mediation analysis has several advantages over frequentist methods, the software tools available have mainly focused on linear structural equation modelling (LSEM). This paper introduces a Bayesian estimation procedure and open-source software tool, BayesGmed, for causal mediation analysis using the Bayesian g-formula approach. The proposed method follows the POF for effect definition and identification and is illustrated using data from a randomized controlled trial.

The paper presents an R-package for conducting causal mediation analysis, which can provide point and interval estimates for causal effects and sensitivity analyses around key assumptions. The proposed methodology was applied to data from a randomized controlled trial on the effects of tCBT on self-perceived change in health status in patients with chronic widespread pain. The results showed that tCBT had a significant effect on reducing sleep problems but none of the potential mediators was found to mediate the effect of tCBT. The paper suggests that improving the scope of tCBT or combining it with other interventions could increase patient benefit and that adequately powered trials are needed to explore these potential mediators and non-specific effects of tCBT.

However, there are some limitations to the package, including the assumption of a parametric specification for the outcome and mediator model, and the consideration of only a single mediator without exposure mediator interaction. The authors plan to extend the package to handle these limitations in a future version, and users can also update the package for their own needs as it is distributed as open-source software.

In summary, this paper is well-organised and solid in the technical part. I would accept it after a minor revision.

I would suggest authors add a section to introduce the related work in mediation analysis since enumerating related work helps to better declare the contribution of this paper.

I list some references for your consider.

1. "Causal mediation analysis with hidden confounders."

2. "Disentangled Representation for Causal Mediation Analysis." 

Both of these are published in recent years and focus on causal mediation analysis.

Reviewer #2: The paper presents a novel Bayesian approach for causal mediation analysis using the Bayesian g-formula and introduces an R-package, BayesGmed, for fitting Bayesian mediation models in R. The authors demonstrate the utility of their approach through a secondary analysis of data from the MUSICIAN study. The paper is well-structured, informative, and provides a comprehensive overview of causal mediation analysis.

Strengths:

1. The authors address a significant gap in causal mediation analysis by proposing a Bayesian approach that overcomes the limitations of frequentist methods, particularly in small sample sizes.

2. The introduction of the BayesGmed R-package is a valuable contribution to the field, providing an open-source tool for researchers and practitioners to use Bayesian causal mediation models.

3. The application of the proposed methodology to the MUSICIAN study data effectively demonstrates the utility of the approach and provides a practical example for readers.

4. The use of informative priors in the probabilistic sensitivity analysis is a useful addition, allowing researchers to assess the robustness of causal identification assumptions.

The comparison of results obtained with BayesGmed and the mediation R-package strengthens the validity of the approach and highlights its potential advantages.

Weaknesses:

1. The paper could benefit from a more detailed discussion of the advantages and disadvantages of Bayesian approaches compared to frequentist methods, providing readers with a better understanding of the implications of their choice.

2. The authors mention that the mediated effects in the MUSICIAN study analysis were not statistically significant using BayesGmed. A more in-depth exploration of why this was the case and how this finding relates to the strengths of the Bayesian approach would be helpful.

3. The paper could provide more guidance on selecting informative priors for the sensitivity analysis and discuss the potential pitfalls and biases introduced by using different priors.

Overall, the paper presents a valuable contribution to the field of causal mediation analysis by proposing a Bayesian approach and introducing an open-source software package for its implementation. While there are areas for improvement, the paper's strengths outweigh its weaknesses and provide a solid foundation for future research and practical applications.

6. PLOS authors have the option to publish the peer review history of their article (what does this mean?). If published, this will include your full peer review and any attached files.

Reviewer #1: No

Reviewer #2: No

---

## [Author Response · Author response to Decision Letter 0]

20 May 2023

Dear Editor, 

Thank you for the valuable feedback and consideration of our paper. We thanks the referees for their valuable comments. We have uploaded a cover letter, response to reviewers, and modified manuscript. 

Thanks,

Belay

---

## [Editor Report · Decision Letter 1]

29 May 2023

BayesGmed: An R-package for Bayesian Causal Mediation Analysis

PONE-D-23-00922R1

Dear Dr. Yimer,

We’re pleased to inform you that your manuscript has been judged scientifically suitable for publication and will be formally accepted for publication once it meets all outstanding technical requirements.

Kind regards,

Debo Cheng

Academic Editor

PLOS ONE
---

## [Editor Report · Acceptance letter]

2 Jun 2023

PONE-D-23-00922R1 

BayesGmed: An R-package for Bayesian Causal Mediation Analysis 

Dear Dr. Yimer:

I'm pleased to inform you that your manuscript has been deemed suitable for publication in PLOS ONE. Congratulations! Your manuscript is now with our production department. 

Kind regards, 

on behalf of

Dr. Debo Cheng 

Academic Editor

PLOS ONE